# A Model of the Early Visual System Based on Parallel Spike-Sequence Detection, Showing Orientation Selectivity

**DOI:** 10.3390/biology10080801

**Published:** 2021-08-19

**Authors:** Alejandro Santos-Mayo, Stephan Moratti, Javier de Echegaray, Gianluca Susi

**Affiliations:** 1Laboratory of Cognitive and Computational Neuroscience, Center for Biomedical Technology, Technical University of Madrid, 28040 Madrid, Spain; alejandro.santos@ctb.upm.es (A.S.-M.); stephan.moratti@ctb.upm.es (S.M.); javier.echegaray@ctb.upm.es (J.d.E.); 2Department of Experimental Psychology, Faculty of Psychology, Complutense University of Madrid, 28040 Madrid, Spain; 3Laboratory of Clinical Neuroscience, Center for Biomedical Technology, Technical University of Madrid, 28040 Madrid, Spain; 4Department of Civil Engineering and Computer Science, University of Rome “Tor Vergata”, 00133 Rome, Italy

**Keywords:** intrinsic excitability, multi-neural spike detector (MNSD), orientation selectivity, spiking neural network simulation, delay-learning

## Abstract

**Simple Summary:**

A computational model of primates’ early visual processing, showing orientation selectivity, is presented. The system importantly integrates two key elements: (1) a neuromorphic spike-decoding structure that considerably resembles the circuitry between layers IV and II/III of the primary visual cortex, both in topology and operation; (2) the plasticity of intrinsic excitability, to embed recent findings about the operation of the same area. The model is proposed as a tool for the analysis and reproduction of the orientation selectivity phenomenon, whose underlying neuronal-level computational mechanisms are today the subject of intense scrutiny. In response to rotated Gabor patches the model is able to exhibit realistic orientation tuning curves and to reproduce responses similar to those found in neurophysiological recordings from the primary visual cortex obtained under the same task, considering different stages of the network. This demonstrates its aptness to capture the mechanisms underlying the evoked response in the primary visual cortex. Our tool is available online, and can be expanded to other experiments using a dedicated software library developed by the authors, to elucidate the computational mechanisms underlying orientation selectivity.

**Abstract:**

Since the first half of the twentieth century, numerous studies have been conducted on how the visual cortex encodes basic image features. One of the hallmarks of basic feature extraction is the phenomenon of orientation selectivity, of which the underlying neuronal-level computational mechanisms remain partially unclear despite being intensively investigated. In this work we present a reduced visual system model (RVSM) of the first level of scene analysis, involving the retina, the lateral geniculate nucleus and the primary visual cortex (V1), showing orientation selectivity. The detection core of the RVSM is the neuromorphic spike-decoding structure MNSD, which is able to learn and recognize parallel spike sequences and considerably resembles the neuronal microcircuits of V1 in both topology and operation. This structure is equipped with plasticity of intrinsic excitability to embed recent findings about V1 operation. The RVSM, which embeds 81 groups of MNSD arranged in 4 oriented columns, is tested using sets of rotated Gabor patches as input. Finally, synthetic visual evoked activity generated by the RVSM is compared with real neurophysiological signal from V1 area: (1) postsynaptic activity of human subjects obtained by magnetoencephalography and (2) spiking activity of macaques obtained by multi-tetrode arrays. The system is implemented using the NEST simulator. The results attest to a good level of resemblance between the model response and real neurophysiological recordings. As the RVSM is available online, and the model parameters can be customized by the user, we propose it as a tool to elucidate the computational mechanisms underlying orientation selectivity.

## 1. Introduction 

The crux of most of the unresolved questions of neuroscience lies in the mechanisms of neuronal cooperation and parallel processing of information in the brain [1]. Although a broad palette of realistic neuronal models is available to today’s computational neuroscientists, there are very few neural circuits, if any, of which the working mechanisms are known with certainty. Nonlinearities, delays and modulations associated to both neuron and synapses, are tough aspects to be addressed when the intent is to characterize neural computation at the level of neural assembly, making the hypotheses on their overall operation difficult to confirm, and the interpretation of the subjacent mechanisms arduous. 

One of the emblems of this deadlock is represented by the open debate on whether neurons code information using firing rates or precise spike timings [2,3,4]. In this regard, combining theory of neural computation with experimental observations, a promiscuous frame emerges. Some biological neural circuitries seem to favor the former strategy, others the latter. Interestingly, while rate coding was previously considered a more plausible and robust strategy than temporal coding, in recent years more and more importance has been given to the significance of the temporal organization of spike patterns [5,6]. Today the rate-based perspective may even appear as an ad-hoc methodological postulate, with no empirical or theoretical support [3], while the temporal coding a plausible and efficient strategy for information representation and transfer [4,7]. Importantly, temporal coding has been associated with computational processes of perception, both alone and in dual action with firing rates [5,8,9,10]. Multi-neuronal spike sequences repetition with millisecond precision, as well as parallel time-structured spike patterns, have been found in brain areas involved in vision, and considered as evident forms of coding [5,11,12,13,14,15].

While on the one hand new technologies allow us to collect accurate knowledge about the structure of real neuronal microcircuits, on the other hand simulators allow us to reproduce them accurately and evaluate if theories are supported by the simulation outcomes. In this way, computational models can help us to test and at least confute hypotheses on their operation. An increasing number of neuro-inspired de/coding structures are being developed with the aim of both facing real world problems and unveiling mechanisms implemented by the brain for information processing in the brain (e.g., [16,17]), having the potential to shed light onto this debate. 

### 1.1. Model Elements

Among the prototypical structures developed for bio-inspired neural microcircuits, the recent multi-neuronal spike sequence detector (MNSD) [18,19] is able to perform online learning and recognition of parallel spike sequences (i.e., sequences of pulses belonging to different neurons/neuronal ensembles). Based on the leaky integrate and fire with latency (LIFL) neuron model [20,21] and neural plasticity, the MNSD combines different learning schemes (weight- and delay-based), and offers a way to understand the neural processing of information. The LIFL is a neuron model whose computational capabilities considerably rely on the neurocomputational feature spike latency (SL) [22]. The latter implies an over-threshold state before the spike generation, and allows the time between threshold crossing and spike generation to vary depending on the membrane potential reached during the summation process. Thereby, spike latency enables to adjust a neuron’s spike time by modulating its internal state. The spike latency mechanism present in the LIFL has been extracted from the biological Hodgkin-Huxley model (see [20]), representing a relevant biological resource to implement temporal coding by many types of neurons.

The intrinsic time-dependent plasticity is a biological concept that has emerged during recent years, which can be considered an internal modulator of the intrinsic neuronal excitability (IE) [23,24,25]. This effect occurs in parallel to the induction of long-term synaptic modifications and produces changes in the neuron’s conductivity, by modulating the h-channel activity. It is defined as a non-synaptic plasticity process because changes are produced within the postsynaptic cell and are not attributed to the strength of the synapse itself. Thus, in presence of intrinsic excitability plasticity, the temporal difference between presynaptic and postsynaptic spike is not only able to produce synaptic long-term potentiation/depression (LTP-s/LTD-s), but also an increase/decrease of the intrinsic excitability (LTP-IE/LTD-IE). This mechanism causes a change in the neuron’s conductance, thus strengthening the modulation of the membrane potential, and finally controlling the spike latency. Many studies have shown that changes in intrinsic excitability contribute to perceptual functions, being found in many sensory brain areas [26,27,28] (e.g., auditory [29], olfactory [30], somatosensory and visual [31] areas). Here, we focus on the primary visual cortex (V1) where mechanisms of intrinsic excitability modulation have also been found [31,32]. The plasticity of the intrinsic excitability has been shown to be regulated by visual experience [26]. Importantly, experimental data from rodents show that this type of plasticity is more expressed in V1 during postnatal critical developmental periods, and then reduced for the rest of life [28,33]. These intrinsic excitability changes might be involved in the tuning of sensory neurons in the visual cortex during critical periods and subsequently in plastic changes of perceptual functions like orientation selectivity in V1.

### 1.2. Rationale under the Model Design

Our model is aimed to reproduce the encoding mechanisms of basic image features, contemplating at the same time multiple neural representations of the information. 

We focused on the computational underpinnings of the first level of scene analysis, which is deputed to detect elementary visual attributes such as line orientation (cap. 25 of [34], by C.D. Gilbert). To do so, our reduced visual system model (RVSM) incorporates the processing chain starting from the retina, passing through the lateral geniculate nucleus (LGN) and arriving at the cortical layers of the V1, involving both temporal- and rate-based signaling (spikes arriving as input to layer IV, and output from pyramidal neurons of the column, respectively). Since both structure and behavior of the MNSD resemble the circuitry among layers II/III and IV of V1 described by [35], we used it as the “detection core” of the RVSM. The input neurons of V1 (i.e., the spiny stellate cells (SS) of layer IV) receive parallel inputs from the LGN/retina, and then converge through excitatory connections to different target pyramidal cells (PC) of layer II/III, arranged in different orientation columns, resulting in a pinwheel-like structure. Basically, when the pinwheel is hit by an input spike pattern encoding the preferred orientation of one of its columns (i.e., angle to which the column is sensitive), the latter will selectively activate.

Critically, important aspects of human cognition like perception have been associated with spike timings [36,37,38], alone or together with firing rates (or spike counts). Specifically, for vision, the existence of both precise spike timings at the retinal and LGN levels and firing rates in orientation sensitive V1 neurons, suggests that synchronization of spike occurrence is needed for the activation of target neurons which are coding orientation. As the two mechanisms are present in the MNSD, our model could be important to elucidate the mechanisms underlying conversion from temporal to rate code as suggested by experimental data and has the potential to unify existing hypotheses regarding early visual processing.

The MNSD used in this work is similar to its original formulation, but it has been adapted to our problem by implementing intrinsic excitability plasticity instead of heterosynaptic plasticity, motivated by recent findings that suggest this mechanism as a plausible biological framework for visual orientation tuning in V1 [32]. This allows for the same computational properties and an even simpler computational scheme. Additionally, we contextualized the MNSD considering the real architecture of V1 layers II/III and IV, i.e., by adding inhibitory populations to each layer and implementing excitatory and inhibitory assemblies of layer V and VI to conform the complete cortical column. In our MNSD-based modules the signal converges to only 4 cortical columns, which architecture is inspired by the work of Potjan and Diesmann [39], forming a pinwheel-like structure which we call reduced pinwheel, able to detect the orientation depicted in the provided images. On the basis of previous work on image coding in V1 [5,40,41], we assumed that image information primarily resides in a short latency window of 20 ms, containing the first spikes that most contribute to recognition performance [40,41]. A scheme of the proposed RVSM is illustrated in Figure 1.

### 1.3. Layout of the Present Work

In the *Materials and Methods* we give a bottom-up presentation of the system components: (1) the neuron model, i.e., Leaky Integrate and Fire with Latency and Intrinsic Excitability (LIFL-IE), which is the basic computational element of our work; (2) the MNSD-based microcolumn, showing how it reproduces the tuning mechanism; (3) the overall RVSM and (4) the methods of signal extraction and comparison implemented. 

We then introduce the overall RVSM model and its performance in discriminating Gabor patches of different orientations, showing that it behaves according to the well-known experiment of Hubel and Wiesel [42] and encompasses the resulting orientation tuning curves. To check the biological plausibility of the model output, electrophysiological-like signal is synthesized from our cortical column when responding to orientated Gabor patches, and then compared with real neurophysiological signal from V1 area under the same task: postsynaptic activity of human subjects obtained through magnetoencephalography (MEG), and spiking activity of macaques obtained through multi-tetrode recordings. The visual evoked response of the model is characterized by amplitude peak at ~90 ms after the stimulus onset, in accord with real human electrophysiological recordings of V1 response to visual stimuli (the so-called C1 component described by Di Russo and colleagues [43]).

Finally, we discuss the conclusions that can be drawn from our results and the limitations of our approach. 

The different RVSM modules can be downloaded at the following link: https://github.com/LCCN/MDPI2021 (accessed on 18 August 2021).

## 2. Materials and Methods

In this section we progressively present the parts that make up our model. First, we present the LIFL-IE spiking/synaptic model. Second, we show the cortical column inspired by the cortical microcircuit proposed by [39] embedding MNSD-based microcircuits along layers II/III and IV. Finally, we show the method used to extract electrophysiological signals from the model, when fed with parallel spike sequences coming from a virtual retina model [44] relayed through the LGN. We implemented the RVSM using NEST simulator [45] (v.2.16), with the tool PyNEST [46], that is a simple and flexible Python interface to design network simulations on NEST. NEST provides the possibility to create new modules with particular types of neurons and synapses, and it is easy to use. In order to carry out the simulations, in this study we implemented the LIFL-IE neuron model in C++, as a NEST extension module (i.e., a library, see Appendix A).

### 2.1. The LIFL-IE Model

#### 2.1.1. The Leaky Integrate and Fire with Latency (LIFL) Neuron

Roughly speaking, neurons are cells characterized by an internal membrane potential (*V_m_*, initially at the resting potential *V_rest_*), able to change in response to input currents and to generate a spike once a threshold (*V_th_*, the threshold potential) is crossed. Neurons are characterized by many neurocomputational features [22], and scientific literature presents different neuron models, each one characterized by its set of neurocomputational features. According to [18], the LIFL neuron model has a fixed spike threshold *V_th_*, and it is able to operate in two different modes: under-threshold (or passive) mode when *V_m_ < V_th_*, and over-threshold (active) mode when *V_m_ > V_th_*. While in the under-threshold mode the neuron behaves as a leaky integrator, in the over-threshold mode it is characterized by the neurocomputational feature spike latency, which consists of a membrane potential-dependent time delay between the overcoming of the potential threshold and the actual spike generation (see [19,22]).

First, we have implemented the LIFL neuron model in the NEST environment. A preliminary adaptation of the original mathematical model’s membrane potential (*V_m,original_*) has been realized, in order to obtain bio-plausible values for the membrane potential of this new implementation (*V_m,NEST_*), i.e., [−70, −54.4] mV on the basis of experimental data [47]. Like in the original LIFL mathematical model, we divided the neuron operation in two modes, i.e., under-threshold (*V_m,NEST_ < V_th_*) and over-threshold (*V_m,NEST_ > V_th_*). Regarding the under-threshold mode, for each integration step, we update the *V_m,NEST_* taking into account the under-threshold decay. Regarding the over-threshold mode, on the basis of the original LIFL neuron model, we compute the *V_m_* envisaging a maximum spike latency of 25 ms between the threshold crossing and the actual spike generation, a value chosen according to [18]. We call this value maximum spike latency (*SL_max_)*, which occurs when an input pulse is able to barely lead the *V_m_* over the threshold. A detailed description of this process, together with mathematical equations involved, is described in Appendix A.

For both modes, every time a pulse arrives to the neuron, it is summed to the current state, taking into consideration exponential synapses in each one of the dendrites. Exponential synapses refer to the shape of the function used to model the conductance change in the post-synaptic neuron resulting from the neurotransmitter release (see Figure 2).

Identical excitatory neurons have been chosen for the realization of the MNSDs, and the input pulses were initially set in order to evoke a spike latency of 12.5 ms in the afferent neuron (Figure 2a). As explained in the Section 2.2.1, during the learning phase of the MNSD the spike latency is modulated by the amplitude of the input spikes (once *V_th_* has been reached, the higher the amplitude of the input, the shorter the presented spike latency).

#### 2.1.2. Plasticity of Intrinsic Neuronal Excitability

The spike-time-dependent plasticity (STDP) has been widely studied [48], in both its homo- and hetero-synaptic forms [49], and implemented in neuromorphic circuits in order to obtain both weight and delay adjustments in response to network activity (see, for example, [50,51] respectively). STDP has been successfully employed in computational models of vision to reproduce orientation selectivity (e.g., [52]). In addition to synaptic plasticity, vision plasticity has been found to involve changes in the intrinsic neuronal excitability [26,27], a non-synaptic plasticity process which concept has emerged recently due to several findings in neurobiology [23,24,51,52,53], and specifically in studies related to V1 [32].

The elementary computational unit of the presented system, that is, the LIFL-IE neuron model, is based on the LIFL neuron and the intrinsic excitability plasticity mechanism. Its inclusion in a patch of V1 cortex, allowed the firing probability of each neuron of the layer II/III to discriminate the activity patterns arriving from the LGN.

As stated by [23,24], the plasticity of intrinsic excitability follows the same rules of STDP. In a nutshell, the input current acting on the synaptic receptors of the postsynaptic dendrites is modulated through the intrinsic excitability parameter (*IE_p_*), implying an increase/decrease of the neuron’s conductance resulting in high/low excitability. *IE_p_* is initially set to 1 and updated for each spike arrival with the following relations:(1)$$IE_{p}=IE_{p,\;previous}+\lambda \cdot e^{\frac{-\Delta t}{\tau }}\;\;,\;\;\;\;for\;\Delta t>0\;$$
(2)$$\;IE_{p}=IE_{p,\;previous}-\lambda \cdot e^{\frac{-\Delta t}{\tau }}\;\;,\;\;\;\;for\;\Delta t<0$$
where, *IE_p,previous_* represents the value of *IE_p_* reached in the previous step; *λ* is the maximum variation of potentiation and depression for each spike arrival; *τ* is the time constant of potentiation and depression (the same positive values for both time constants and amplitudes are chosen for simplicity for potentiation and depression); and Δ*t* (defined as *t_pre_-t_post_*) the time difference between the presynaptic pulse arrival and the postsynaptic spike generation (see Figure 2b). Thus, a neuron that undergoes intrinsic excitability potentiation will reach a higher membrane potential with the same input pulse.

A dedicated NEST module of LIFL neuron (i.e., lifl_psc_exp_ie, see Appendix A) contemplating this plasticity effect through the ‘soma_exc’ parameter of the neuron model, has been implemented. The plasticity of this intrinsic excitability is obtained by previously defining some presynaptic modulators as ‘stimulator’ of the postsynaptic neuron. According to [24], we set the IE temporal windows (*τ*) in the range of tens of ms (12.5 ms in our case).

### 2.2. The Multi Neuronal Spike-Sequence Detector (MNSD)

#### 2.2.1. MNSD Based on LIFL-IE

Several studies suggest that in perceptual processes precise spike patterns are involved in the handling of environmental information (e.g., [5]). For this reason, the basic assumption of our model is that V1 uses spike pattern detectors to decode the incoming visual information. Consequently, we implemented in our model a spike pattern detector, the MNSD, which has both functional and architectural analogies with the circuitry comprised between layers II/III and IV [54].

Regarding the architectural aspects, layer IV receives input from magnocellular and parvocellular pathways, and converges information to few neurons of layer II. At this point, the horizontal connections between the delay neurons of our model (abundant in the layers of the V1 visual cortex, see [34]) allows the model to recognize multi neuronal spike patterns related to a column’s preferred orientation, that are finally signaled by a readout neuron. The MNSD architecture (composed here of 4 pulse generators and 5 neurons, see Figure 3) is able to learn and then discriminate between different spike sequences through the spiking activity of one of the neurons, called target neuron. After a learning phase presenting the same multi-neuronal spike sequence (encoding, in our case, a particular orientation bar), only this pattern will cause the target neuron to generate a spike.

The MNSD is articulated in three layers:*Pulse generator layer*. This layer is composed of 4 pulse generators (i.e., *E*_1_ to *E*_4_) that mimic external inputs arriving from the LGN. The generators are set to produce a specific spike sequence. Referring to Figure 3, the pattern is characterized by four pulses delayed by 5 ms each.*Delay layer*. This layer is composed of four LIFL neurons (i.e., the delay neurons *D*_1_ to *D*_4_) that exploit the spike latency and intrinsic excitability plasticity for the operation of the structure. They receive the spike sequence using one-to-one connections with the previous layer (*E_n_ → D_n_*) and project to the downstream target neuron; in addition each delay neuron is connected to its neighbors, acting as modulators (in the MNSD we implemented in this work the first and last neurons are connected in a “circular” way). These connections are characterized by a small weight, which is necessary to obtain the intrinsic excitability plasticity effect, but not so strong as to be a relevant contribution to the membrane potential.*Target layer*. Here we exploit the leaky behavior of the LIFL neuron model. The target neuron T receives the spikes of the four delay neurons; each one will cause an initial increase of *V_m_*, followed by a decay toward *V_rest_*, until the arrival of a new incoming spike. We set the weights to the target neuron in order to make it produce a spike (overcoming of *V_th_*) only when the spikes arriving from the delay layer are sufficiently synchronized (*simultaneity condition*).

Through the amplitude-time transformation operated by the spike latency feature, it is possible to obtain synchronization at the target neuron acting on the amplitude of the pulses at the input of the four delay neurons. Synchronization at the target neuron in response to a specific input sequence is progressively obtained through the repeated presentation of such sequence to the structure, due to the interplay between the spike latency and the plasticity of intrinsic excitability.

As shown in Figure 3b (left), the target neuron does not produce a spike due to the lack of synchrony between the arriving pulses (if *V_m_* < *V_th_*, then the target neuron remains in the under-threshold mode). During the learning phase of the MNSD the spike latencies of the neurons of the delay layer are modulated acting on the amplitude of their input: the greater the *V_m_* evoked in the postsynaptic neuron, the smaller the latency of the produced spike, and vice versa. The same input pattern is represented after training, but this time the target neuron reaches the threshold and generates a spike (see Figure 3b, right) due to the synchronization achieved among the output spikes of the delay layer. To obtain such synchronization, we repeatedly present the same spike pattern to the structure. During this process the intrinsic excitability of each delay neuron is self-adjusted resulting in the desired spike latencies. Once the learning is completed, the target neuron will generate a spike only when this specific spike pattern is presented.

For the implementation we choose a step of 0.1 ms to obtain a fair trade-off between quick and accurate simulations. The same pattern is repeatedly presented 300 times. Each trial lasts 1 s with the intrinsic excitability plasticity effect activated (NEST model parameter stdp_mod set to true). Once the learning process was completed, we checked the architecture with different spike sequences maintaining the intrinsic excitability plasticity switched off (NEST model parameter stdp_mod set to false). To this purpose, we changed the spike order of the input sequence (possible combinations) and performed the recognition.

The extension module developed for this paper can be downloaded from our GitHub repository, installed in NEST using the CMake installation guide (“NEST developer space”) and then used in Python with the command nest.Install(‘LIFL_IE’). The values of neuron parameters are reported in the Table 1. In addition to the extension module, in the same GitHub folder the user can download the Python script (MNSD_with_LIFL_IE.py) to simulate the simple MNSD shown in Figure 4.

#### 2.2.2. Spike Sequence Detection Using the MNSD

As described above, the input current of a certain delay neuron is multiplied by its intrinsic excitability parameter (*IE_p_*) to produce variations on the *V_m_*, which translates into variations of the spike latency. In Figure 4a,b, we can see the difference between the behavior of the structure to the same input pattern before and after 300 learning trials. In addition to be successful in pattern learning, the detector must show specificity when other spike patterns are presented, i.e., the detector should not only make the target neuron spike for the learned patterns, but also avoid spiking for alternative patterns. In order to test network specificity we presented different input sequences to the structure. 

As we can see in Figure 4d, the value of intrinsic excitability for each neuron can be greater or smaller than 1. Figure 4e displays how the differences between spike arrival times decrease throughout the presentation of the learning trials, leading to a minimum time difference between them, leading the structure to elicit a spike in the target neuron only for the presented spike pattern.

Since the changes of spike times generated by the delay neurons is mediated by the changes in the intrinsic excitability, we used the NEST multimeter to monitor the intrinsic excitability value during the simulations, by recording the variable ‘soma_exc’.

### 2.3. Visual Cortex Orientation Tuning Reproduction Using a MNSD-Based Architecture

The brain analyzes visual scenes at three levels [34], of which the lowest one is dedicated to detect simpler visual attributes such as local contrast, color and orientation. In this section we explain how the model reproduces the mechanism of orientation tuning and discrimination, pertaining to the lower level of scene analysis, realizing a sensing associated to de/coding circuitry which contemplates both classical and novel vision principles. We show its ability to learn input patterns represented in the temporal domain, and selectively respond to them as found in V1 [13]. 

To show the resemblance of our model with respect to the V1 neocortical circuitry, we first synthesized the RVSM and then characterized its orientation tuning curves after training. For the sake of simplicity, we synthesized a reduced model of the early visual processing, without considering the difference of the responses due to the contra- and ipsi-lateral visual pathways. This is because we are focusing on the orientation decoding mechanism itself, that seems to be independent of the optic nerve crossing which happen in the real visual complex [34]. Additionally, we schematize the layer IV sublayers (IVCα and IVCβ) as a single layer, considering a whole input from magnocellular and parvocellular pathways.

In nature the primary visual cortex is characterized by columnar structures formed by orientation sensitive cells that show an increment of activity to a preferred orientation. Consequently, orientation tuning functions of V1 neurons show a generalization gradient fitting a gaussian curve.

In the proposed RVSM, the retina consists in the COREM model [44]. We presented Gabor patches (18 by 18 pixels) as visual input to the retina which gives rise to 324 input current values, each mapped to a ganglionar cell. Then, the first spikes of the ganglionar cells were extracted to form the visual spike pattern representing the Gabor patch. The temporal structure of this first wave of ganglionar spikes reasonably carries all the information needed for recognition [13]. This process was repeated for 37 Gabor patches rotated from 0° to 180° at steps of 5°. The model has been then sized considering real data from vision literature [55,56,57], as summarized in Table 2 and Table 3 and detailed in Appendix B.

In this adaptation of the MNSD, we have a 4-delay-neurons implementation of the MNSD in V1, where LGN afferences are working as spike generators, SS cells as delay neurons, and PCs as target neurons (as represented in Figure 3). All cortical column layers respond to visual stimuli but layers II/III modulate the output with higher activity in deep layers (V and VI) when the oriented pattern is presented as found in [58]. This proposed columnar model can be found distributed around each V1 pinwheel detecting the 180° orientations [59]. In this work, for simplicity, we have used 4 cortical columns as previously described, each one trained to detect one of the 4 orientations characterizing the reduced-pinwheel.

The simulation, which scripts can be found in our Github repository, has been divided in two phases: training and test. During the training, we presented to one of the orientation columns of the pinwheel the same spike patterns (corresponding to its preferred orientation). This has been achieved by giving the retina with a Gabor patch 500 times, with 200 ms intervals per trial to get the column selective to this orientation. This process causes each SS neuron of layer IV of the specific column to adapt through the intrinsic excitability parameter to detect the stimulated orientation (i.e., 0°). It is repeated with the three resting columns, using 45°, 90° and 135° Gabor patches. Each column is trained for a specific orientation and the plasticity of intrinsic excitability of our SS neurons was switched off for the test phase.

Considering one single trained column, the test phase consisted of presenting each of the 37 oriented Gabor 30 times. Once we tested the responsiveness to an orientation by the single column, we extracted the tuning curve, which is a measure of how activity response depends on the orientation of a stimulus. To achieve this we calculated the firing rate, i.e., the number of spikes produced by layer II/III PC neurons in a short time of windows, 50 ms to 100 ms. PC activation responses after training were characterized by orientation tuning curves with peaks at the preferred (trained) Gabor patch orientation. The orientation selectivity index (OSI) is a common measurement of cortical columns response in orientation tasks broadly used in vision literature. Specifically, OSI is a measure of how a cell’s firing rate depends on the orientation of a stimulus, defined as:(3)$$OSI=\frac{R_{pref}-R_{orth}}{R_{pref}+R_{orth}}$$
where *R_pref_* and *R_orth_* are the firing rates at the preferred (*−*45°) and orthogonal (45°) orientations, respectively for each pyramidal layer. This index offers us a measure to understand how our layer II/III PC neurons are working in our model.

The simulator allows us to monitor the activity and electrical variables of each neurons of the cortical oriented column over time (e.g., spike raster plot, membrane potential, postsynaptic current) through the so-called multimeters. 

### 2.4. Comparison with Experimental Data

Regarding MEG data, thirty healthy right handed volunteers (22 females, 8 males. age = 23.9 ± 4) participated in the study after giving written informed consent. The signal was recorded during the consecutive presentation of 240 Gabor patches stimulating C1 response in right V1 [43] The study was carried out at the UCM-UPM Laboratory of Cognitive and Computational Neuroscience (Center for Biomedical Technology, Madrid), with full ethical approval from the local ethics committee. Recording was performed using an Elekta 306 channels MEG device placed in a shielded room, with 600 Hz sample rate. The participants underwent passive viewing of the 240 Gabor stimuli (gray-shaded Gabor patch subtending 1.44 visual degrees). Noise artifacts derived from eyeblinks and cardiac activity were corrected by independent component analysis while 300 ms before stimulus onset were used for the baseline correction for each trial. Additionally, signal for each trial was filtered by 0.5 to 60 Hz band-pass before average. Linearly constrained minimum variance source estimation was computed, using the overlapping spheres forward model [60,61] to estimate current dipole activity from right upper calcarine vertex obtained for each trial. All MEG data pre-processing and source reconstruction were performed using the Brainstorm toolbox [62]. To extract from the model a signal corresponding to those recorded with MEG, we calculated the simulated postsynaptic current response, (since both derive from the same postsynaptic signal [63]), by integrating the postsynaptic currents that reach the apical dendrites of the PC. Due to differences in the contribution of PC to the current dipole moment, the total integrated postsynaptic activity was multiplied by the average apical dendrite length to obtain more realistic measures, giving more relevance to layer V and VI PC, to emulate the MEG signal generation process [63] (length of apical dendrites for PC have been chosen as in [64]). Thus, multiplying postsynaptic activity (in pA) of layers II/III, V and VI by 4.82 × 10^−4^ m, 1.342 × 10^−3^ m and 9.63 × 10^−4^ m respectively, we obtained the current moment dipole in pAm unit.

Regarding spiking activity, recordings of two adult *Macaca Mulatta* from the V1 gratings dataset of Bethge Lab (University of Tübingen, Tübingen, Germany; see [65] and the data availability section) were used. The dataset concerns 24 custom-built tetrodes recordings of spike data, across 17 sessions, while 45° sine gratings at 100% contrast visual stimuli were presented to the macaques. We obtained a quantity analogous to the multi-unit-activity (MUA) from both real data and the RVSM by summing up the spike number of all neurons along 0 to 110 ms in bins of 10 ms and then performing signal smoothing. Since our model is based on human V1, a lag between model’s and monkey’s peaks was expected, with anticipated response in macaque in accord with experimental data (e.g., [66]). To estimate this lag, a cross-correlation between monkey and RVSM spike activity was computed. Resulting peak was taken as reference for the temporal shifting of monkey signals to match with RVSM output. Finally, similarity between real and simulated signal was estimated by Pearson correlation coefficient between 25 ms and 100 ms for both postsynaptic (human MEG signal vs. RVSM postsynaptic signal) and spiking (monkey MUA vs. RVSM MUA) activity.

## 3. Results

We first tested the performance of the RVSM in recognizing different Gabor patch orientations. To cover one orientation cycle (i.e., the 180° angle) four oriented cortical columns composing the reduced pinwheel structure had been previously trained (see Section 2) and set their preferred orientations to 0° (or 180°), 45° (or 225°), 90° (or 270°) and 135° (or 315°). For simplicity, along the text we refer to these columns as column 0, column 45, column 90 and column 135, respectively.

We assess the ability of the RVSM in reproducing (1) the orientation tuning curves of the pinwheel cortical columns, and (2) the shape of the electrophysiological signal when responding to orientated Gabor patches.

### 3.1. Orientation Tuning Showed by the Columnar Structures

Here, we gave static inputs to the retina and observed the output of the layer II/III of a single column (i.e., the target neurons of the MNSDs). Specifically, we presented for 200 ms a Gabor patch characterized by a specific orientation and observed the evoked firing rate on the pyramidal layer II/III (target neurons) designated to detect that specific orientation; subsequently, we repeated the experiment with patches of different orientation. As Hubel and Wiesel’s early experiments on V1, we observed a pronounced response for one particular orientation (i.e., the preferred orientation of the oriented column) and weaker responses for similar orientations, falling off significantly when the line orientation is about 40 degrees away. Importantly, the obtained generalization gradients resemble a Gaussian shape, as in the real case. The behavior of the model is illustrated in Figure 5. 

Regarding the orientation tuning showed by the whole pinwheel, we presented to the model images with angles ranging from 0° to 180°, and measured the activity produced by the four oriented columns of the pinwheel, obtaining a behavior that resembles that of the real case (described in [67]). Specifically, a set of Gabor patches of 37 progressive orientations covering the 180° was presented for 30 times (i.e., 30 trials), and finally the firing rate of the cortical layer II/III during the visual evoked response (50 ms to 100 ms) has been measured.

Orientation selectivity of our model has been finally quantified using the OSI index of layer II/III [68] along the 30 trials obtaining an averaged OSI of 0.72 (S.D. = 0.17).

An oriented stimulus will generate an increase of spikes at a specific detector column (i.e., the one that has been trained with this orientation). This implies an increment of the postsynaptic response amplitude compared with the other columns of the pinwheel. A multi-level illustration of this mechanism is given in Figure 6, where the same Gabor patch (45°) is contemporarily presented to a cortical column trained to detect that specific orientation, and to a cortical column trained for the related orthogonal orientation. In the first case, the synchronous spike activity of SS neurons of layer IV generates the detection of the specific orientation and triggers the activity of the rest of the layers, while in the second case SS neurons fire asynchronously, being unable to trigger the neural activity of subsequent layer VI. This reflects in a stronger postsynaptic response produced by the column designated to detect the specific orientation of the stimulus.

### 3.2. Comparison between the Synthetic and Real Evoked Activity

Here we evaluate the similarity between the visual evoked response of our RVSM with that of humans and rhesus monkeys exposed to the same stimulus (see Figure 7).

First, we compared the simulated postsynaptic current from the whole pinwheel with source-space MEG activity of V1. Since dipole moment activity of the RVSM is computed from all simulated pyramidal neurons and real V1 activity extracted from source-space MEG signal, both signals have the same unit of measurement (pA·m).

Next, we compare the spiking activity of our whole RVSM model to the real spike activity of the macaca recordings. Note that scale factor differs from each signal due to the different number of neurons involved in each process. 

Possible lags between model and the experimental data (human and monkey activity) are estimated through thee cross-correlation. In Figure 7a the comparison between simulated pinwheel response and the real V1 activity at postsynaptic and spiking level is illustrated. 

Regarding postsynaptic signal (Figure 7b, top), both the curves have a rise phase starting at approximately 60 ms, and a peak at approximately 80 ms. The Pearson correlation coefficient was estimated among 25 and 100 ms to test their similarity in the interval of interest (r = 0.96; *p* < 0.001). The cross-correlation (Figure 7c, top) indicates a near to 0 lag (peak at −2 ms) between the two signals. Analogously to MEG signal, similarity across spiking activity of monkey shifted signal and RVSM among 25 and 100 was estimated through the Pearson correlation coefficient (r = 0.86; *p* < 0.001). In spiking activity, cross-correlation between monkey and RVSM model shows a relevant latency peak (8.33 ms, Figure 7c, bottom). This lag observed with monkey is in accord with [66], where visual activity in V1 is found at around 60 ms for the macaque (i.e., earlier than the human V1 response).

## 4. Discussion

Nature offers incredibly effective and efficient models that can be used as an inspiration for innovation, and this often motivates the designers of engineering systems to rely on bio-inspired solutions. A big number of computer vision systems are based on neuromorphic structures, then progresses in reverse engineering the human vision system [69] would result not only in an improvement of existing approaches, but also to test hypotheses about the neurocomputational mechanisms of brain areas such as V1 and the orientation selectivity property.

Recent works show that the first wave of spikes in the inferotemporal cortex around 100 ms after image presentation carries enough information for object recognition indicating the importance of early spikes [70]. Many neurophysiological and computational evidences about the importance of first-spike coding [71,72] confirm that information in the brain is transferred by spike patterns with dedicated structures to code and decode stimuli [13,73]. 

Here, we present a system aimed at reproducing early V1, based on the neuro-inspired MNSD structure [18], which has been properly contextualized it in the visual orientation tuning mechanism based on recent findings on visual cortex. This area shows plasticity of intrinsic neuronal excitability [23,74,75], which has shown to be relevant in visual perception training [26], particularly for layer IV neurons [76,77]. The existence of postsynaptic changes of the intrinsic excitability in V1 [26,27,32] suggests a role in experience-dependent refinement of visual cortical circuits [28]. This intrinsic excitability modification has been found in rodent visual cortex only throughout the critical period [33,78] indicating the natural predisposition to train visual perception in these occipital cortices after birth. Thus, although plasticity of intrinsic excitability is reduced in this brain area once the critical period is over, orientation discrimination is maintained. This biological process could be understood as the training stage underlying the tuning of the orientation curves, allowing the network to perform its perception role throughout the rest of the rodent life, although short term plasticity is also implicated during learning [79,80].

Then, to provide a realistic framework for the MNSD operation, we embedded the mechanism underlying plasticity of the intrinsic excitability into the neuron itself instead of synaptic plasticity. Our model can learn (and consequently detect) parallel spike sequences. As can be seen in the left side of Figure 6, only specific time-locked spike patterns (coming from layer IV SS cells) are able to trigger the cascade of pyramidal activation leading to the visual evoked response. 

After training, the architecture is able to successfully reproduce the typical orientation tuning curves found in neurons of the V1 [81], showing a physiological low error rate, due to (1) the small number of neurons involved in our architecture (less than 1000 neurons per column), which reflects an underrepresentation of the pinwheel and visual field and (2) the Poissonian noise adopted in our model (used to confer realism to the RVSM). Even though, an OSI value of 0.72 (layer II/III) was reached by the model indicating high discrimination between orientations, a value that encompasses the experimental findings [82,83]. However the absence of random noise would lead to higher OSI values but rendering the proposed RVSM more artificial than the real V1.

The present work aimed for a bridge between the spiking neural networks and the electrophysiological neuroimaging using the second as a tool to further test the model operation. For this purpose, postsynaptic activity from the RVSM pinwheel was computed as a product of the cortical column response to the presented visual stimuli. The visual evoked response is well-known and commonly used in neuroimaging to assess first visual processing in the cortex. For the experimental data used, Gabor patch stimuli was presented to elicit a clear visual C1 component, i.e., the first cortical activation originated in V1 [42]. Biologically, the C1 component is formed by a characteristic waveform whose peak is around 80–90 ms. Similarly, the activity of the pyramidal cells composing our simulated cortical columns, reproduced both waveform and latencies observed in early human visual processing of V1. The simulated signal found resemblance with both human and monkey recordings.

Recent literature reflects a marked interest of the scientific community towards computational models of orientation selectivity in the visual cortex. Among the relevant works, Ursino and La Cara [84] proposed three simple models as practical tools to test hypotheses on the disposition of cortical synapses avoiding massive computational efforts. From the comparison between simulations and experimental data they found a pronounced plausibility for one of the tested models (the so-called antiphase inhibition model), at least for cat recordings, claiming that the same principle be operative in other animals (such as ferrets or primates). Despite the encouraging results, the authors underline among the limitations of their study the fact that temporal aspects have not been incorporated in the model, representing a central subject for subsequent extensions of the studies in this field. Interestingly, Chariker and colleagues [85] developed a model of the V1 to reconcile its visual functions on the typically sparse LGN input. Their model is not only capable of emulating experimental data merely in the orientation selectivity mechanism, but involves other interesting aspects like diversity in neuronal response and oscillations. From their model emerges that intracortical interactions play a major role in all aspects of the visual functions. Among the conclusions, they recommend to giving emphasis to population dynamics and, where possible, moving toward data-driven, comprehensive models. Nguyen and Freeman [86] recently developed a model to explain how the inputs to cortex come to be spatially segregated, supporting a mechanism in which the segregation occurs through Hebbian strengthening and weakening of geniculocortical synapses during the development of the visual system. 

In addition to the fact that our model contemplates at the same time multiple neural representations of the information, it brings together the aspects highlighted by the cited works as it involves temporal aspects and is able to clearly show how population dynamics affect the overall model behavior. Importantly, it is oriented toward a data-driven, comprehensive modeling of the mechanisms underlying orientation selectivity, including the changes in the intrinsic excitability as form of plasticity which modulates the architecture of the visual system during its development. Taken together, these elements highlight the usefulness of our model in elucidating the mechanisms underlying orientation selectivity. Considering that it supports conversion from temporal to rate code, as plausibly suggested by experimental data, it has the potential to unify existing hypotheses regarding early visual processing. 

## 5. Conclusions

We propose a system aimed at reproducing early V1, based on the neuro-inspired MNSD structure and supporting the conversion from temporal to rate code as suggested by experimental data. The model has been applied to perform a perceptual task obtaining a behavior that resembles that of the real case.

The developed tool offers a high customization level, enabling the researcher to quickly and easily explore configurations of interest. We adapted the plasticity of intrinsic excitability and spike latency features to an existing neuron model (through C++) to use NEST as a simulation environment, providing a tool for further large-scale MNSD-based pattern detection systems. To conclude, we point out that among the positive aspects of NEST, there is the possibility of simulating different neuron models, and to parallelize the computation [87,88], considerably reducing simulation times. The NEST module allowed us to adequately synthesize the RVSM under an efficient simulation framework, to simulate a cognitive task in which spike timing is an essential feature, which could be used in future works to model other sensory cortices or cognitive functions. 

## Figures and Tables

**Figure 1 biology-10-00801-f001:**
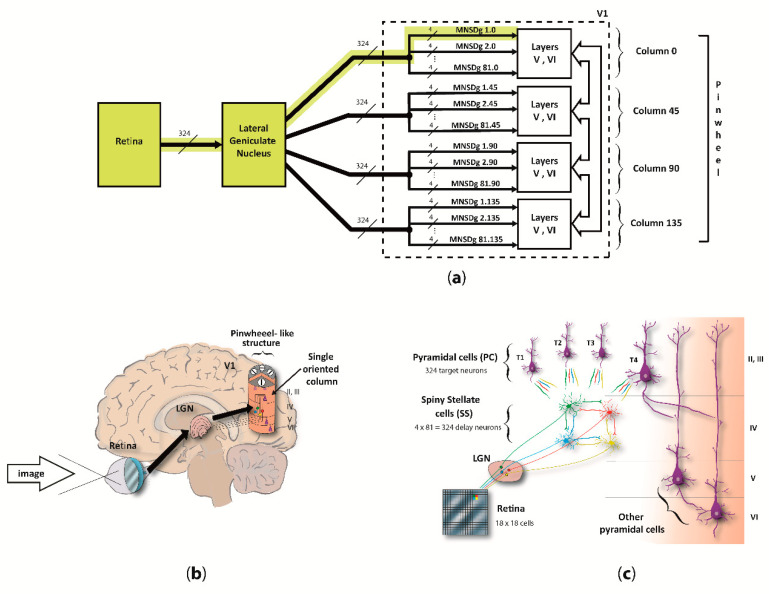
Illustration of the RVSM.: (**a**) Wiring scheme of the complete system, from the 324 cells of the retina to the MNSD-based microcircuits residing in layers II/III and IV (connection cardinality is indicated along the tracts). Each one of the 4 columns (i.e., *column 0*, *column 45*, *column 90* and *column 135*) is composed of 81 MNSD groups, and indicated with the notation *ID _MNSD group_.ID _column_* (e.g., MNSDg 2.45 means the 2nd MNSD group of cortical column 45). Neurons of layers V and VI are shared among the MNSD structures; (**b**) Anatomical contextualization of the neuronal path highlighted in (**a**): from stimulus presentation to one of the four V1 oriented columns composing the *pinwheel-like* structure; (**c**) Diagram of neuronal connections from a specific section of the retina (represented by 4 neighboring cells out of the 324) to a single MNSD-based microcircuit of one of the four cortical columns (delay neurons of layer IV to target neurons of layer II/III). The output connections to layers V and VI are shown for one of the targets only. For ease of understanding, in (**b**) we depicted only the MNSD structure formed with one of the targets, although each set of four delay neurons (layer IV) is connected to four target neurons (levels II/III) giving rise to four different MNSD structures, i.e., a MNSD group (as specified in (**c**)). The reader can find a precise description of the network in the Section Materials and Methods and in Appendix B.

**Figure 2 biology-10-00801-f002:**
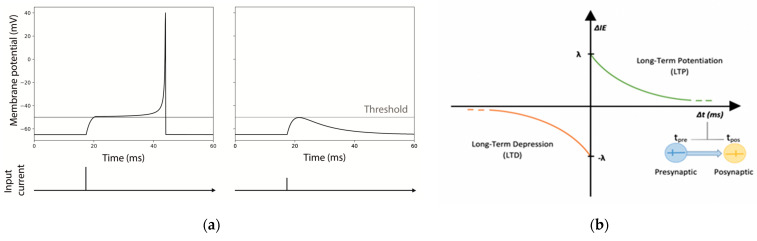
Behavior of the LIFL-IE module: (**a**) internal computation of LIFL-IE under an impulsive excitatory injected current. The first input is sufficiently strong to lead the *V_m,NEST_* over-threshold and generate an action potential, while the second input is too weak to lead the *V_m,NEST_* over the threshold. The vertical line represents the stimulus onset, whereas the horizontal one represents *V_m_*; (**b**) scheme of intrinsic excitability plasticity, showing IE-LTP and IE-LTD. Δ*t* is the time difference between presynaptic postsynaptic spikes.

**Figure 3 biology-10-00801-f003:**
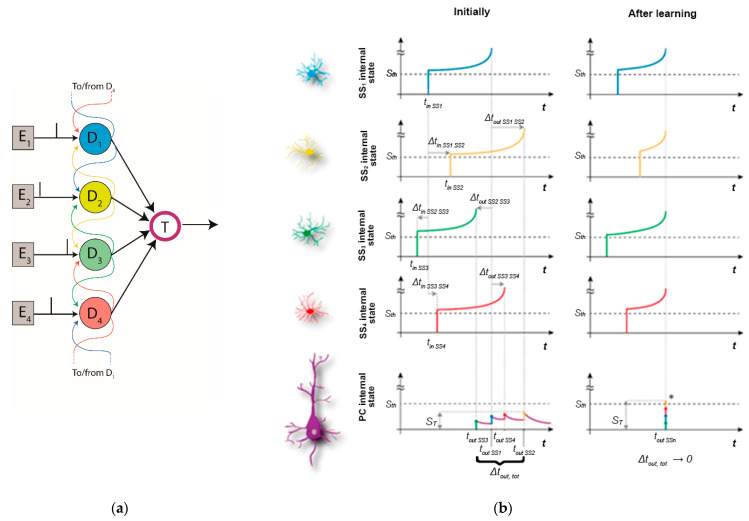
MNSD behavior. (**a**) Diagram of the MNSD architecture. Each circle represents a neuron and the gray boxes represent the input pulse generators. (**b**) Diagram of the operating principle of the MNSD with 4 branches. Δ*t_in Dm,Dn_* represents the time difference between the pulses afferent to the delay neurons (i.e., *t_in Dn_* − *t_in Dm_*), and Δ*t_out Dm, Dn_* the time difference between the pulses afferent to the target neuron (i.e., *t_out Dn_*
*− t_out Dm_*). On the left, desynchronized input pulses are unable to activate the target; this is because delay neurons fire asynchronously, and the target neuron does not fire due to desynchronization between the arriving spikes (where Δ*t_out, tot_* represents the total time interval occupied by the multi-neuronal spike sequence arriving to the target neuron). On the right (after training), thanks to the spike latency modification the delay neurons’ output spikes are generated synchronously (*simultaneity condition:* Δ*t_out, tot_ →* 0) enabling the target activation. Finally, the maximum internal state reached by the target neuron (S_T_) is represented for both the cases.

**Figure 4 biology-10-00801-f004:**
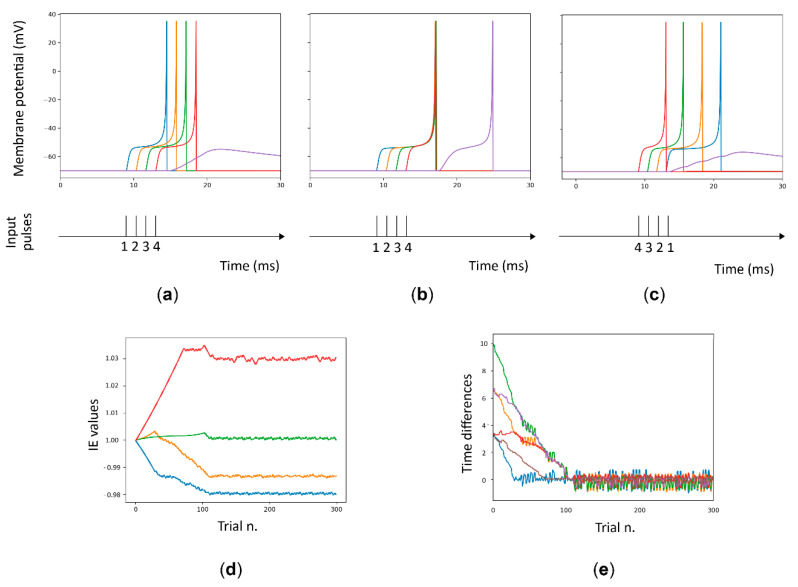
MNSD learning phase. Plot of the training patterns before (**a**) and then (**b**) the MNSD is trained; in the second case the target (red line) generates a spike. (**c**) the trained MNSD is now stimulated with the same pattern timings but different order. This does not allow the target to reach the threshold. Plot of the MNSD training: (**d**) trajectory of the intrinsic excitability parameter (*IE_p_*) for each delay neuron and (**e**) trajectory of the time difference between two neuron spikes in ms (Δ*t_out_*). In the latter figure, during the presentation of the trials, the values progressively tend to zero, indicating that neurons fire together.

**Figure 5 biology-10-00801-f005:**
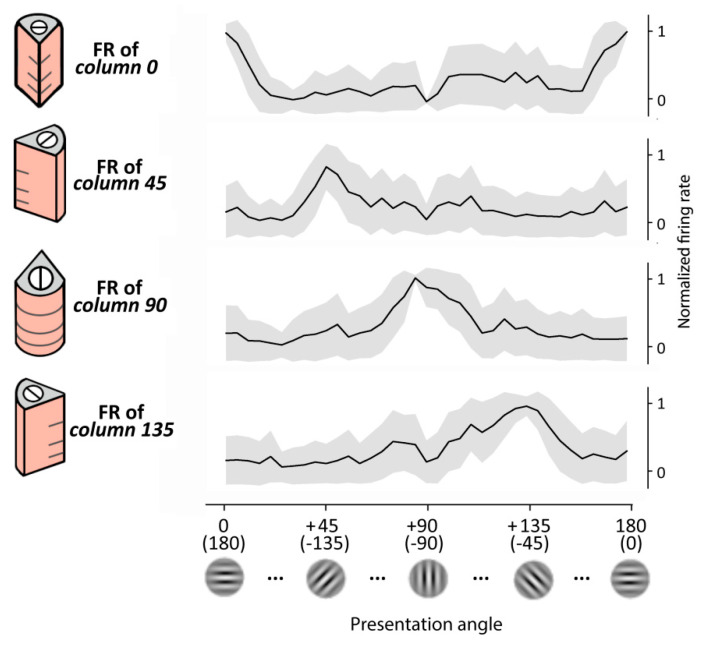
Orientation selectivity of each cortical column of the reduced pinwheel of the RVSM. A set of 37 Gabor patches were presented to the model for 30 trials. Each of the four subplots represents the firing rate (FR) of layer II/III pyramidal neurons of the column indicated in the y-axis after the presentation of the Gabor patches with orientation indicated in the x-axis. The more the angle of the presented stimulus is similar to the preferred orientation of the column, the higher the FR generated by the latter. The values obtained from each set of presentations have been normalized to the maximum FR obtained in each column. For a comparison with typical orientation tuning curves of neurons of the V1, see, e.g., [67].

**Figure 6 biology-10-00801-f006:**
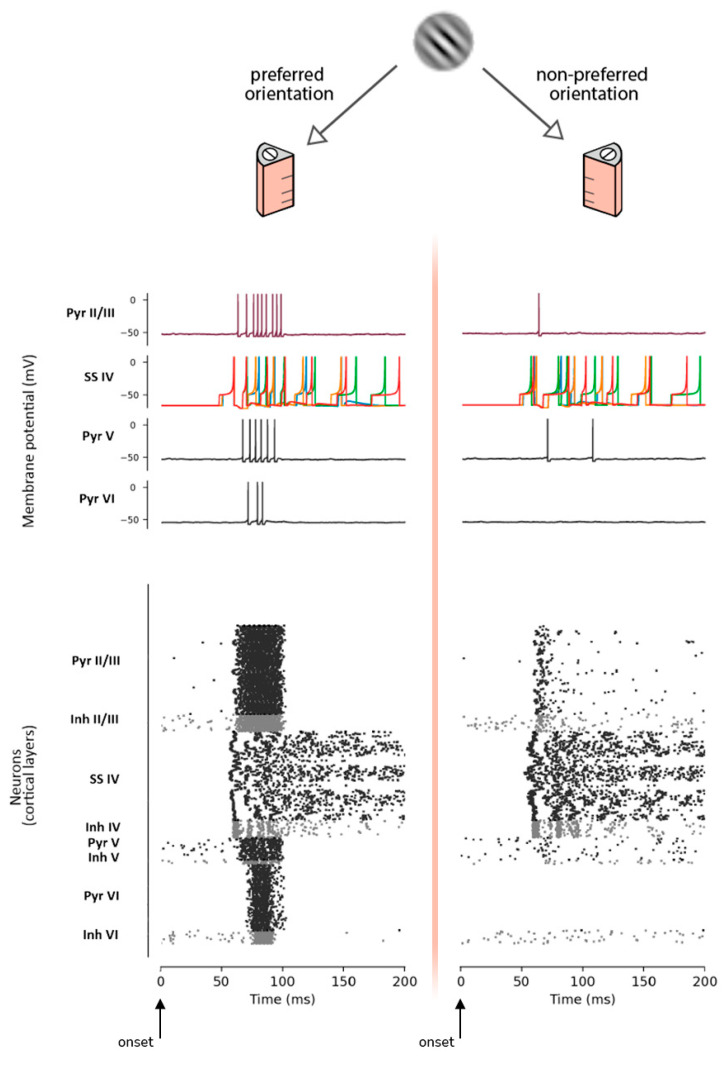
Response to the same Gabor patch (45°) operated by a cortical column with that preferred orientation (i.e., *column 45*, left) and a cortical column trained for the related orthogonal orientation (i.e., *column 135*, right). Top: scheme of the presentation. Middle: plot of the membrane potential of a MNSD structure (target pyramidal neuron of layer II/III, related 4 SS neurons of layer IV) and single pyramidal neurons of layers V and VI. Bottom: raster plot of the whole oriented column showing the firing activity (excitatory in black and inhibitory in gray).

**Figure 7 biology-10-00801-f007:**
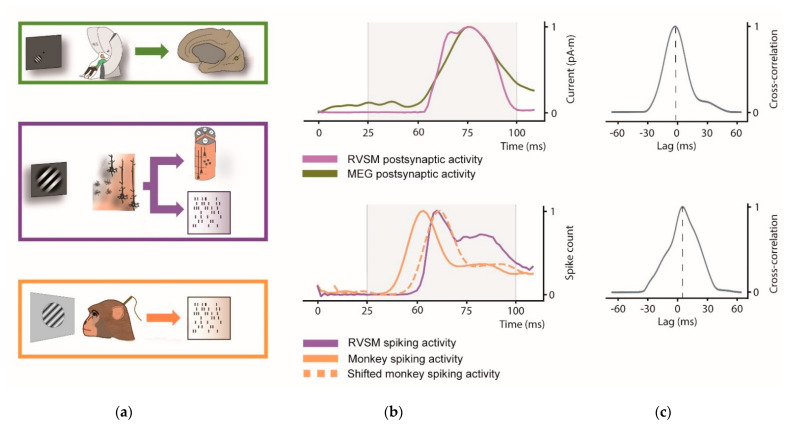
Response time series of the visual evoked potential obtained after the presentation of Gabor patch in human, RVSM model and monkey V1. (**a**) Scheme of the pipeline for real and simulated data. Activity arising from a left lower Gabor patch visual stimulation was recorded by MEG and then source reconstruction was computed to extract current activity of upper right V1 (green). The same Gabor patch was presented to the artificial retina, computing all pinwheel columns responses in order to obtain the current activity from apical dendrites of pyramidal neurons and spiking activity of all pinwheel neurons (purple). Rhesus spiking activity of visual evoked response recording to the same orientation and contrast Gabor patch viewing (orange); (**b**) Top: MEG signal averaged on the 30 subjects (green) and current dipole activity of the complete pinwheel of the RVSM responding to a 45° Gabor patch (purple). Bottom: spiking activity of RVSM (purple), monkey response (orange) and lag-corrected monkey response (dashes orange). Both plots were normalized by scaling between 0 and 1 to span the same range. Shifted activity for MEG recordings is not reports since it is considered *near 0-lag*; (**c**) Cross-correlation of postsynaptic (top) and spiking (bottom) activity to estimate the lag between the real and model activity.

**Table 1 biology-10-00801-t001:** Most important neuron parameters of ‘lifl_psc_exp_ie’ model for the NEST extension module *‘LIFL_IE’*, used to simulate the simple MNSD.

Neuron Parameter	Description	Value
‘decay’	LIFL-IE underthreshold decay	0.02
‘stimulator’	ID of the presynaptic modulator neurons	--
‘lambda’	intrinsic excitatory maximum amplitude	0.0005
‘tau’	intrinsic plasticity window	12.5
‘stdp_mod’	Switch on/off the spike dependent modification of intrinsic excitability	‘true’

**Table 2 biology-10-00801-t002:** Network size and neuron types of the computational model implemented. The term layer is here abbreviated with *L*.

	L II/III	L IV	L V	L VI
**Population Size**	PC: 324 Inhibitory: 65	SS: 324 Inhibitory: 65	PC: 81 Inhibitory: 16	PC: 243 Inhibitory: 49
**Model (excit.)**	‘aeif_psc_exp_peak’	‘lifl_psc_exp_ie’	‘aeif_psc_exp_peak’	‘aeif_psc_exp_peak’
**Model (inhib.)**	‘aeif_psc_exp_peak’	‘aeif_psc_exp_peak’	‘aeif_psc_exp_peak’	‘aeif_psc_exp_peak’
**Layer inputs**	L IV	LGN	L II/III	L IV

**Table 3 biology-10-00801-t003:** Connectivity parameters of the computational model implemented (indegree per postsynaptic neuron; weight = 100 pA, delay = 1 ms). The term layer is here abbreviated with *L*.

			FROM							
			L II/III		L IV		L V		L VI	
			PC	inh	SS	inh	PC	inh	PC	inh
**TO**	**I/III**	**PC**	36	8	4	-	-	-	-	-
		**Inh**	35	8	-	-	-	-	-	-
	**L IV**	**SS**	-	-	3	6	-	-	-	-
		**Inh**	-	-	32	6	-	-	-	-
	**L V**	**PC**	15	-	-	-	10	8	-	-
		**Inh**	-	-	-	-	30	8	-	-
	**L VI**	**PC**	-	-	-	-	20	-	20	6
		**Inh**	-	-	-	-	-	-	32	6

## Data Availability

The source code and data used to produce the results and analyses presented in this manuscript are available from the Github repository: https://github.com/LCCN/MDPI2021 (accessed on 18 August 2021). The V1 gratings dataset (Bethge Lab) is available at the following link under a CC-BY license: http://bethgelab.org/datasets/v1gratings/ (accessed on 18 August 2021).

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
