# Peer review of "A Model of the Early Visual System Based on Parallel Spike-Sequence Detection, Showing Orientation Selectivity"

_biology, 2021, doi:10.3390/biology10080801_

Round 1

Reviewer 1 Report

Overall, the manuscript is well-written and it is nicely presented for using a reduced visual system model (RVSM) as well as multi-neuronal spike sequence detector (MNSD) system to depict orientation coding in primary visual cortex combining both computational method and experimental data. And I’m convinced that this work highlights the significance of converting from temporal to rate code and the potential to unify existing hypotheses regarding early visual processing.

I have only few minor comments.

  1. First of all, the numbers used in Figure 1 should be explained, such as in Figure 1a, 1c. 2.
  2. there is clearly a typo in Figure 1a, where should be ‘Retina’.
  3. Besides, in line 108, it should be ‘but not at the presynaptic site’ when authors said ‘and not at the synapse site’?
  4. Line 472, ‘Noise artefacts’, artifacts?
  5. An dashed line indicate stimuli onset could be nicer in Figure 6.

Author Response

Dear reviewer,

thank you for your kind comments and useful suggestions. We have fixed the points you raised:

1. In the caption of Fig.1 we gave an explanation of the numbers involved in the schemes (appearing in both a) and c) ).

2. We corrected the typo in "retina".

3. We modified line 108 trying to be clearer on the concept we mean, i.e., that the changes are produced within the postsynaptic cell and are not attributed to the strength of the synapse itself.

4. "Artefacts" has been substituted with "artifact". 

5. To make the diagram of Fig 6 easier to understand, we replaced the dashed separator using a pink bar, and explicitly indicated the onset below the time axes.

Thank you and best regards.

Reviewer 2 Report

This manuscript proposes an artificial visual system based on MNSDs, which show orientation selectivity after learning and have biological similarities with the real visual system. This research is of reference significance for the reconstruction of the visual system. The paper contents are abundant, and the narration of the article is organized. Some contents that might help further improving the quality of this paper are listed as follows:

  1. Some issues with several figures.

Figure 1 (a): ‘Retina’;

Figure 5: +90(-90);

Appendix: A1-A4 are blurred. The author would better not use screenshots to show calculations and formulas.

2.The design scheme of RVSM is not explained clearly.

Although the model is given in Figure 1 (a), the details of the inputs’ allocation rules are very few, and the clear microcircuits connecting all MNSDs and all target neurons are not mentioned in the paper. I suggest the author give more description about how the 324 input is assigned to each MNSD and explain whether one input is only received by one MNSD and whether one target neuron is only connected to four MNSDs.

3.The introduction to the input pattern is omitted.

According to Figure 1 (c), it seems that the grayscale Gabor patches are used as training data, but there is not so much discussion about the relations between pixel values (from the retina) and spike pattern (spike sequence E1- E4). Besides, the description of the processing flow (from input to output) in this manuscript is brief. It is strongly suggested to give detailed processing flow in a local receptive field (retina -> MNSD -> target neuron). In short, to give more information about how the grayscale value inputs cause a specific spike pattern.  Furthermore, the relations between Vm, IE, and SL are described in words and are very brief, and the author may give more details about their relations (such as formula or calculation rules).

  1. Some minor problems.

“simultaneity condition: Δtout, tot → 0” , what is the meaning of ‘tot’? (Target output time?)

According to Figure 1(c), it seems that D1 also should ‘connect’ to D4 (Figure 3 (a))?

Author Response

Dear reviewer, 

thank you very much for the motivating words and the proposed corrections

POINT 1:

Issues related to figures 1 and 5 have been fixed.

The formulas in appendix have been rewritten with the proper equation tool.

POINTS 2 and 3:

Thank you for the observations. We realized that actually the operation of the RVSM information routing had to be explained in more detail in order to understand the operation more clearly. Then, we described in Appendix B how the information is tranformed along the path.

POINT 4:

"tot" stays for "total", referred to the time interval occupied by the spike train arriving to the target neuron. We specified it in the caption accordingly.

Finally, thank you for notifying us the error: In the text we now specified that, in the MNSD we implemented here, the first and last neurons are connected in a “circular” way, and the figure has been modified accordingly.

Thank you and best regards

Reviewer 3 Report

The model shows to replicate the neural response and orientation tuning properties of the target neurons. However, it is not clear what the significance of the model is beyond validating the sufficiency of some neurophysiological mechanisms, such as IE, for reproducing certain neural behaviors, orientation tuning here. For example, how predictive the model is? i.e., is the model able to predict other oriented stimulus patterns beyond the Gabor used for the model training?

The model profits from biologically plausible assumptions on the model architecture or parameters based on the literature, but it does not sound that the model will add to the current understanding of possible mechanisms underlying the V1 orientation tuning. Are there any other implications/applications that the authors envision for perhaps more engineering (e.g., computer vision applications or improving SNNs to reach or surpass the ANNs' performance for a similar discrimination task) rather than possible scientific impacts of this work?

Author Response

Dear reviewer,

We would like to thank you for the valuable comments and constructive suggestions given for the improvement of our work.

The model is aimed at elucidating the computational mechanisms underlying orientation selectivity, which are today partially unclear. Specifically, considering the current understanding of possible mechanisms for the V1 orientation tuning, we support and highlight the importance of the conversion from temporal to rate code, which is corroborated by experimental data. To our knowledge, our model is the first to support the idea of a conversion from temporal to rate code, an aspect that has the potential to unify existing hypotheses regarding early visual processing.

Your final question is very interesting for us, as we plan for future work to extend this study to real application. Our group is not just oriented to a bio-realistic model of early vision system, but also to engineering applications of SNNs synthesized for recognition tasks. As an extension of this work together with [1] (where we carry out simulation batteries using real world images), we aim to optimize in the future our tools for computer vision applications. Among the main points to be addressed, we envisage:

  • to develop a GUI which makes possible a more user-friendly interaction (i.e., make the size of the input block fully-settable by the user, set the number of columns per pinwheel, etc.);
  • to include 2 additional blocks, i.e., image-segmentation (to manage the division in visual fields), and a bridge to make it compatible with available classification tools.

Thank you and best regards.

[1] - Susi G., Antón-Toro L.F., Maestú F., Pereda E., Mirasso C.. "nMNSD—A Spiking Neuron-Based Classifier That Combines Weight-Adjustment and Delay-Shift". Front Neurosci, 2021.